# *Staphylococcus epidermidis* and *Cutibacterium acnes*: Two Major Sentinels of Skin Microbiota and the Influence of Cosmetics

**DOI:** 10.3390/microorganisms8111752

**Published:** 2020-11-07

**Authors:** Mathilde Fournière, Thomas Latire, Djouhar Souak, Marc G. J. Feuilloley, Gilles Bedoux

**Affiliations:** 1Laboratoire de Biotechnologie et Chimie Marines LBCM EA 3884, IUEM, Université Bretagne Sud, 56000 Vannes, France; tlatire@uco.fr (T.L.); gilles.bedoux@univ-ubs.fr (G.B.); 2Laboratoire de Biotechnologie et Chimie Marines LBCM EA 3884, IUEM, Université Catholique de l’Ouest Bretagne Nord, 22200 Guingamp, France; 3Laboratoire de Microbiologie Signaux et Microenvironment LMSM EA4312, Université de Rouen Normandie, 27000 Évreux, France; djouhar.souak@univ-rouen.fr (D.S.); marc.feuilloley@univ-rouen.fr (M.G.J.F.); 4BASF Beauty Care Solutions France SAS, 69007 Lyon, France

**Keywords:** skin microbiota, biofilm, cosmetic, *Staphylococcus epidermidis*, *Cutibacterium acnes*

## Abstract

Dermatological and cosmetics fields have recently started to focus on the human skin microbiome and microbiota, since the skin microbiota is involved in the health and dysbiosis of the skin ecosystem. Amongst the skin microorganisms, *Staphylococcus epidermidis* and *Cutibacterium acnes*, both commensal bacteria, appear as skin microbiota sentinels. These sentinels have a key role in the skin ecosystem since they protect and prevent microbiota disequilibrium by fighting pathogens and participate in skin homeostasis through the production of beneficial bacterial metabolites. These bacteria adapt to changing skin microenvironments and can shift to being opportunistic pathogens, forming biofilms, and thus are involved in common skin dysbiosis, such as acne or atopic dermatitis. The current evaluation methods for cosmetic active ingredient development are discussed targeting these two sentinels with their assets and limits. After identification of these objectives, research of the active cosmetic ingredients and products that maintain and promote these commensal metabolisms, or reduce their pathogenic forms, are now the new challenges of the skincare industry in correlation with the constant development of adapted evaluation methods.

## 1. Introduction

Research on human microbiota in dermatology started in the 1950’s with Kligman, with the improvement in cell culture [1]. In 2008, the National Institutes of Health launched the five-year Human Microbiome Project with the objective to sequence human microbiome components. Emergence of new technologies, such as next-generation sequencing (NGS), enable the comprehensive study of the human microbiome [2,3]. The microbiome corresponds to the set of genomes of microorganisms in symbiosis with the human host, including microorganisms, alive or dead, and free DNA, whereas microbiota refers to microorganisms living in or on a defined ecosystem [4]. Human microbiota, mainly established on the skin, oral and vaginal mucosa, as well as in the respiratory, urinary and gastrointestinal tracts, have fundamental roles in health and diseases [5,6]. The human adult skin microbiota is comprised of diverse microorganisms, including bacteria, fungi [7,8], yeasts, viruses [9], archaea [10] and mites, principally *Demodex* [11].

The skin, the largest organ of the human body, is a complex and dynamic ecosystem. Its outer epidermal layer, the stratum corneum, is the first physical barrier that prevents chemical substances or pathogenic microorganisms’ entrance, fluid evaporation and body heat loss [12]. The stratum corneum is composed of 75–80% proteins, mainly keratins and membrane proteins; 5–15% lipids; and 5–10% unidentified compounds [13,14]. For the establishment of this microbiota, the skin provides essential nutrients, such as amino acids from the hydrolysis of proteins, fatty acids from the stratum corneum, sweat, lipid hydrolysis or sebum and lactic acids from sweat [15]. Microorganisms are based on this stratum corneum and on the deeper cutaneous appendages [16,17]. The skin microbiota, made up of millions of commensal microorganisms, e.g., 1 million/cm^2^, is the second largest microbiota of the human body in mass [18]. Cutaneous bacteria belong to four main phyla among the thirty-six known [17]. The average skin body distribution of these main bacteria phyla, detected on 20 diverse skin sites of 10 healthy individuals, were found to be Actinobacteria at 51.8%, Firmicutes at 24.4%, Proteobacteria at 16.5% and Bacteroidetes at 6.3% [19,20]. Procurement of the skin microbiota occurs in the early stage of birth. In utero, the skin is sterile, devoid of any microorganism, and is colonized a few minutes after birth by commensal microorganisms of the mother, depending on the childbirth method [21,22,23]. This colonization process in the neonatal stage is essential for the establishment of immune tolerance towards commensal microorganisms [1,24]. Microbiota colonization continues during growth until reaching an equilibrium state in adulthood [1].

An increase in scientific investigations into skin microbiota has inevitably led to the emergence of related studies in the cosmetic industry [25], which has now become unavoidable in the cosmetic market. Indeed, skin microbiota is involved in the maintenance of a healthy cutaneous barrier. Skin and microorganisms live in symbiosis and microorganisms help to maintain the skin barrier, the immune system and limit pathogenic microorganism growth [26]. However, an imbalance in skin microbiota, called dysbiosis, is correlated with skin pathological diseases, such as acne with the loss of phylotype diversity of *Cutibacterium acnes* and atopic dermatitis with the increase in pathogenic *Staphylococcus aureus* and commensal *Staphylococcus epidermidis*, as well as with non-pathological diseases, such as sensitive and dry skins [4].

The principal challenges of such research are (1) the comprehension of each microorganism’s role in healthy and beautiful skin; (2) the description of the microbiota of weakened skin with the precise determined cause or consequence of the correlation between dysbiosis and skin pathologies; and (3) evaluation studies of the active ingredients’ effect on skin microbiota. The maintenance, protection and restoration of microbiota diversity and equilibrium, as well as prevention of skin dysbiosis, are the emerging claims of cosmetic products. These products may include prebiotics, probiotics, post-biotics or active ingredients with demonstrated effects.

This review highlights the fundamental role of *S. epidermidis* and *C. acnes* in the skin system as two skin microbiota sentinels, since they can protect against and prevent pathogens, as well as participate in skin equilibrium in their commensal form with the secretion of beneficial metabolites. However, according to skin microenvironment perturbation, their metabolism can be modulated, and their pathogenic form can disturb skin microbiota homeostasis with biofilm organization. Therefore, the current evaluation methods for cosmetic purposes, targeting skin microbiota and particularly the two sentinels *S. epidermidis* and *C. acnes*, considering both their assets and limits, are discussed. Finally, the mode of action of commercialized active cosmetic ingredients for the maintenance of these two beneficial skin sentinels are finally highlighted in this review.

## 2. Interactions between Skin, *Staphylococcus epidermidis* and *Cutibacterium acnes*: A Possible Shift from Commensalism to Opportunistic Pathogenicity

### 2.1. Staphylococcus epidermidis and Cutibacterium acnes: Two Major Commensal Gram-Positive Bacteria of Skin Microbiota

*Cutibacterium* and *Staphylococcus*, two genera of stable Gram-positive bacteria (G+), are fundamental components of skin microbiota and highly distributed across the human body according to skin environmental conditions, such as temperature, ranging between 31.8 and 36.6 °C, and pH, ranging between 4.2 and 7.9—the two main factors affecting their growth (see Figure 1) [6]. *Staphylococcus* species are mainly found in moist areas, gathering in axillary vaults, antecubital creases, popliteal creases, and plantar tissues. *Cutibacterium* species are mostly found in sebaceous areas, such as the face (glabella, alar crease and external auditory canal) and the back (see Figure 1) [6,17,27].

*Cutibacterium acnes*, from the Actinobacteria phylum, is an anaerobic, aerotolerant, diphtheroid, bacillus-shaped bacteria, with an optimal growth temperature of 37 °C [4,28]. *C. acnes*, formerly named *Propionibacterium acnes* [29], consists of six main phylotypes, namely, IA1, IA2, IB, IC, II and III [30,31], and ten principal ribotypes, RT 1 to 10 [32,33], encoded in unique rDNA 16S alleles. Due to its anaerobic growth, *C. acnes* is mostly found deeper in the epidermal layer, especially in pilosebaceous units (>95%) [32], within hair follicles, hair and sebaceous glands [32,34]. *C. acnes* growth areas correspond to environments rich in oleic and palmitic acids from sebum known as a sebaceous area (see Figure 1) [20]. The bacteria metabolizes fatty acids and other sebaceous fluids or lipids to generate, through its lipase activity and via the core metabolism, free fatty acids (FFAs) such as propionic and acetic acids [35].

*Staphylococcus epidermidis*, from the Firmicutes phylum, is static, relatively hydrophobic, aero-anaerobic facultative and organized in clusters, with an optimal growth temperature between 30 and 37 °C. *S. epidermidis* differs from other *Staphylococcus* strains by the absence of coagulase activity. *S. epidermidis* is at the level of the stratum corneum and can be found in the epidermal basement membrane in dry, moist and sebaceous regions (see Figure 1) [36,37]. Polysaccharide Intercellular Adhesin (PIA), a linear polymer of N-acetylglucosamine, is the major constituent of the extracellular matrix of *S. epidermidis*.

*S. epidermidis* and *C. acnes* are usually considered as commensal bacteria because they are harmless in healthy conditions and benefits the skin while it is unaffected [38]. These bacteria, through their microbial surface components recognizing the adhesive matrix molecules, interact with the human skin proteins of the skin extracellular matrix, such as dermatan sulphate or type I collagen [39,40], and with the toll-like receptors (TLR2 and TLR4), which then allow the coordination pathways between the bacteria and skin cells [41]. More precisely, commensal *S. epidermidis* and *C. acnes* have a symbiotic relationship with the cutaneous system and more specifically a mutualistic one, since both organisms find benefit [38]. The skin hosts *S. epidermidis* and *C. acnes* and supplements the bacteria with nutrients, while in turn both bacteria participate in skin homeostasis, host defense and innate immunity (see the right part of Figure 2) [38,41].

In healthy skin microbiota and equilibrated skin conditions, *S. epidermidis* and *C. acnes* help the skin in fighting pathogenic bacteria via several mechanisms.

The main mechanism established by *S. epidermidis* and *C. acnes* is the direct inhibition of pathogen growth. These commensal bacteria fight against pathogen invasion due to the competition for nutrients and occupation of an ecological niche by both strains. *S. epidermidis* and *C. acnes* can also desensitize the TLRs by continuous exposure to these commensals or decrease TLR expression [4].

Other tools of these two commensals to directly kill pathogens is the production of bacteriocins and the induction of the production of antimicrobial peptides (AMPs) by host cells, such as keratinocytes, sebocytes and immune cells [41,42]. AMPs are effector molecules of the innate immune skin system, mainly cathelicidins and human beta defensins, hBDs [43].

Indeed, *S. epidermidis* is able to inhibit the adhesion of the virulent state *S. aureus* and pathogenic strains of *S. epidermidis* forming biofilms [46]. *S. epidermidis* prevent their biofilm formation and destroy biofilm via secretion of bacteriocins, produced under the control of its quorum-sensing (QS) system [47], metalloproteinases (MMPs), phenol-soluble modulins (PSMs) and endopeptidase serine protease (Esp) (see Figure 2) [48,49]. PSMs have similar structures to human AMPs with the ability to strongly interact with the lipidic microbial membrane and cause leaks [50]. Bacteriocins produced by commensal *C. acnes* strains have in vitro an action directed against virulent pathogenic strains of *Cutibacterium*, lactic acid bacteria, Gram-negative bacteria and yeasts. The authors suggest that this process might be thus in vivo protect the pilosebaceous unit and the secretory duct against pathogenic microorganisms [38,45]. Via a TLR2-dependent mechanism, keratinocytes synthetize AMPs (especially hBD2 and hBD3) under the influence of *S. epidermidis* and its secretion of molecules with a molecular weight inferior to 10 kDa (see Figure 2) [51]. In healthy skin, *C. acnes* is also able to inhibit the growth of methicillin-resistant *S. aureus* strains. Indeed, *C. acnes* ferments glycerol into short chain FFAs, mainly propionic acid, which lowers the environmental pH, and thus protects pilosebaceous follicles from pathogen growth, such as *S. aureus* (see Figure 2) [52], and promote lipophilic yeast growth. *S. epidermidis*, via the fermentation of glycerol into scFFAs such as acetic, butyric, lactic and succinic acids, inhibits biofilm formation of *C. acnes* [53].

The other mechanism in which *S. epidermidis* acts is host innate immunity enhancement. After induced-AMP secretion, the NF-кΒ pathway is activated and an increase of pro-inflammatory cytokines levels in the epidermis occurs [51,54,55]. Activation of TLR2 by commensal *S. epidermidis* also increases the tight junction barrier in keratinocytes in vitro culture, which contributes to the preservation of epithelial barrier homeostasis [56]. The lipoteichoic acid (LTA) of *S. epidermidis* can also inhibit skin inflammation during wound healing or in skin pathologies via a hypothesized crosstalk mechanism mediated by TLR2 and TLR3 [57]. This might involve an increased induction of mastocyte cells recruitment in contaminated skin regions, or a stimulation of T lymphocytes maturation via TLR2 activation [58], or a reduction of *C. acnes*-induced skin inflammation via the inhibition of TLR2 expression [59]. For the latest mechanism, LTA from *S. epidermidis* activates TLR2 and induces microRNA-143 (miR-143) in keratinocytes. In turn, miR-143 decreases the stability of TLR2 mRNA and its protein production, which inhibits *C. acnes*-induced pro-inflammatory cytokines IL-6 and TNF-α [59]. Succinic acid from *S. epidermidis* inhibits the surface TLRs of keratinocytes and suppresses *C. acnes*-induced IL-6 and TNF-α [60,61]. All this data supports the idea that the TLR2 of different skin cell types, such as keratinocytes, sebocytes and immune cells, is highly involved in *C. acnes* recognition and inflammation initiation in acne. Commensal *S. epidermidis* appears as a potential new player in the physiopathology of acne [44,62].

Promotion of commensal growth *S. epidermidis* and *C. acnes*, and their metabolites or by-product production, such as bacteriocins, propionic acid, LTA or small molecules <10 kDa, are interesting pathways to investigate for cosmetic purpose in pathogen prevention and maintenance of skin microbiota homeostasis.

### 2.2. Staphylococcus epidermidis and Cutibacterium acnes: Shift to an Opportunistic Pathogenicity and Correlation with Common Skin Dysbiosis

It is noteworthy that when talking about skin microbiota, it is not “all white or all black” and shifts between commensalism and pathogenicity are commonly observed when the skin barrier is disturbed (drastic increase of skin pH, water loss, skin flaking, keratinocytes apoptosis and inflammation). In skin pathogenic conditions, virulent *C. acnes* and *S. epidermidis* strains are commonly found under their biofilm forms, producing critical virulence factors, and correlated to skin dysbiosis such as acne [63] or atopic dermatitis (AD) [64], respectively. As a reminder, dysbiosis is a disequilibrium of microbiota diversity and functionality, often characterized by an abnormality in bacterial composition, abundance or deficiency, and correlated to the development of inflammatory skin diseases according to the severity of the dysbiosis [1]. However, the link between cause and consequence between microbiota dysbiosis and skin pathogenesis still remains unclear and is not well established to date [17]. Skin bacterial biofilm is a bacterial community with bacteria tightly attached to a biotic surface—the human skin [63]—and embedded in a complex matrix rich in polysaccharides, produced by the microorganisms, which protects bacteria growth in a hostile environment. Biofilm formation is influenced by the physico-chemistry of the surface (moisture, pH and temperature), nutrient availability, host receptors and local immune system [63].

#### 2.2.1. *Staphylococcus epidermidis* Biofilm and Loss in *Staphylococcus aureus* and *Staphylococcus epidermidis* Diversity: Involvement in Atopic Dermatitis

While focusing on skin area, opportunistic *S. epidermidis* strains were detected in patients suffering from atopic dermatitis [65], the most common multifactorial inflammatory skin pathogenesis [66]. AD is characterized by skin barrier dysfunction with (1) deficiency in various lipids, such as ceramides, cholesterol and free fatty acids; (2) deficiency in keratinocyte protein differentiation, such as filaggrin, loricin and involucrin; (3) immune system stimulus with released pro-inflammatory cytokines after T-cells activation; (4) decrease in antimicrobial peptide (AMP) production by skin cells targeting bacteria; and (5) microbiota alteration with a decreased bacterial diversity [67]. *S. epidermidis* is predominant in patients with less severe AD disease while *S. aureus* is predominant in more severe AD disease [68]. *S. aureus* colonization increase is largely correlated with AD with up to 90% of this strain in patients with AD [66]. Although, *S. epidermidis*, the other predominant species, is present up to 20%. Multi-species staphylococcal biofilms of *S. aureus* and *S. epidermidis* are predominant in this altered microbiota in the form of multi-layered biofilms between the stratum corneum, sebaceous glands and hair follicles [69]. Both *S. aureus*, with 100% of the isolates, and *S. epidermidis*, with 75% of the isolates from AD lesions, are strong biofilm producers, according to their absorbance value in an in vitro biofilm detection [65,70]. *S. epidermidis* biofilm adhesion and formation [71,72,73,74,75,76], and its QS with an *agr* (accessory gene regulator) system [77,78,79] and eventually LuxS system [79,80], have been extensively reviewed and will not be detailed in this manuscript by the authors. These biofilms might form, in AD lesional skin, sweat duct occlusions due to the presence of water and salt, and thus influence the severity of the AD pathogenesis [64,70]. The role of *S. aureus* and potentially *S. epidermidis* as causative in AD needs to be determined.

#### 2.2.2. *Cutibacterium acnes* Biofilm and Loss in Phylotype Diversity: Involvement in Acne Dysbiosis

Acne vulgaris is a common skin chronic inflammatory disease of the pilosebaceous unit. Three main actors are involved in acne development: (1) sebaceous glands with qualitative and quantitative hyperseborrhea [81,82,83]; (2) bacterial strains *C. acnes* and *S. epidermidis* both present in acne lesions, with *C. acnes* representing less than 2% of the skin surface bacteria and *S. epidermidis* being over-represented [32,45,84]; and (3) keratinocytes with hyperkeratinization of the pilosebaceous unit, leading to comedones, papules and pustule formations [85].

Acne is not due to a significant difference in relative abundance of *C. acnes* between healthy and acneic patients but is linked to a loss in *C. acnes* phylotype diversity with a predominance of phylotype IA1 in the pathology from 55% to 68% of all isolates [86,87,88]. In inflammatory acneic lesions, the proportion of *C. acnes* phylotype IA increases while phylotypes IB and II decrease. Phylotype IA or IA1 strains from acneic patients produce larger levels of virulence factors, such as triacylglycerol lipase [89], porphyrins [90,91,92], hyaluronate lyase [93] or Christie–Atkins–Munch–Petersen (CAMP) [94,95] than other healthy phylotypes. These various virulence factors can also interact with molecular oxygen, generating reactive oxygen species (ROS) and free radicals that in turn damage the keratinocytes and thus support perifollicular inflammation in acne [95,96,97]. The authors suggest that this process of keratinocytes and extracellular matrix damage through *C. acnes* virulence could be extended in the skin ageing pathway and could later particularly interest the cosmetic industry. Metagenomic analysis showed that ribotypes RT4 and RT5 of phylotype IA are associated with acne while RT6 is found in healthy skin [32]. Virulent expression of acneic and non-acneic *C. acnes* strains are associated with different environmental niches. The RT4 acneic strain, with its lipophilic surface, exhibits maximal growth and biofilm formation activity in Sebum-Like Medium (SLM), which mimics the sebaceous gland environment. In contrast, the phylotype II RT6 commensal strain, with its more polar surface, preferentially grows in Reinforced-Clostridial Medium (RCM), a medium enriched in oligo-elements and amino acids, and has maximal biofilm activity in Brain Heart Infusion (BHI), the classical medium. RT4 exerts higher inflammatory potential on keratinocytes than RT6, independently of culture medium, which confirm that this ribotype contributes to inflammation. Proteome analysis of the strains revealed that the virulence factors expression was strain-specific and medium-specific. Indeed, the CAMP factor was overexpressed in BHI and RCM by the RT4 strain while triacylglycerol lipase was expressed in SLM only by RT6 [98].

It has been demonstrated in 2007 that *C. acnes* is able to form biofilms in vitro [99], and more recently that its biofilm formation is influenced by temperature with an optimum of 37 °C [100]. In 2012, it was showed, for the first time, that *C. acnes* phylotypes IA and II were able to grow in macrocolonies by producing an important biofilm deeply in sebaceous follicles [84]. Biofilm of these *C. acnes* virulent phylotypes is a key virulence factor in acne pathogenesis [32,99,101,102,103]. Indeed, *C. acnes* isolates from acneic skin are in vitro stronger biofilm producers than isolates from healthy skin [102]. In addition, biofilm formation ability generally correlates with phylotype since a microtiter plate assay showed that phylotype IA1 skin-derived isolates exhibited the highest biofilm formation, followed by phylotypes IC, IA2 and II, while the IB and III phylotypes exhibited the lowest biofilm formation [104]. Its biofilm contributes to the production of exopolysaccharide biological glue that holds corneocytes together and leads to the apparition of microcomedones [105].

The promotion of acne inflammation is performed by secreted virulence factors and biofilm of phylotype IA, especially the RT4 and RT5 strains (see Figure 2). The detailed mechanism of skin inflammation induced by virulent *C. acnes* in acne pathogenesis has been extensively reviewed by Lee et al. (2019) [106]. Briefly, virulent *C. acnes* type IA in acne pathogenesis activates mainly TLR2 in keratinocytes, sebocytes, monocytes and macrophages, which induce an inflammatory response with activation of the NF-κB pathway and produce ROS. The induced *C. acnes* inflammation in acne is characterized in vitro by the production of pro-inflammatory cytokines, such as interleukins and TNF-α; AMPs, mainly hBD2; and MMPs, such as MMP-2 and -9; all in human dermal fibroblasts and sebocytes monoculture [107,108,109]. Virulent *C. acnes* also stimulates sebum production [97] and induces abnormal differentiation and proliferation of epidermis cells [110,111]. The *C. acnes* phylotype IA1 strain, in comparison with the combination of phylotypes IA1 + II + III, cultivated in vitro in a healthy skin explant from abdominoplasties, significantly upregulates the innate immune markers’ expression (interleukins, TLR2 and hBD2), as well as the pro-inflammatory cytokines’ supernatant level [87]. This result on a 3D skin model reinforces the data of previous studies on cell monoculture.

Biofilms, virulence factors and QS of *S. epidermidis* and *C. acnes* appear as interesting targets for cosmetic purposes since both these bacteria under pathogenic form are involved in common skin pathogenesis. Their biofilm formation is mainly conditioned by the skin microenvironment with which these two commensals/opportunistic pathogen strains are in constant contact.

#### 2.2.3. Influence of Skin Microenvironment on *Staphylococcus epidermidis* and *Cutibacterium acnes* Virulence and Biofilm Formation

The communication system between microbiota and skin can be divided into (1) host stress mediators, which are either neurotransmitters, neuropeptides or neurohormones; (2) host immune markers, such as cytokines and AMPs; and (3) microbial factors from bacterial origin. These communication molecules, mediated via the endocrine system involving sweat and sebum or via direct contact with the epidermis and dermis [112], are able to condition biofilm formation and virulence pathways of *S. epidermidis* and *C. acnes* in their pathogenic form or to coordinate positive responses between the skin cells and these bacteria.

Skin, the largest neuroendocrine organ of the human body, secretes several host stress mediators such as substance P (SP), calcitonin gene-related peptide (CGRP), catecholamines and natriuretic peptides (NUPs) [113]. This leads to the emerging concept of cutaneous bacterial endocrinology that must be considered in dermatology [114]. These host stress mediators influence *S. epidermidis* and *C. acnes* metabolism.

SP and CGRP directly control commensal *S. epidermidis* virulence and biofilm formation. Both SP and CGRP, synthetized by Merkel cells, released by free cutaneous nerve endings and colocalized at the class C sensitive fiber level, are known for their involvement in neurogenic skin inflammation [115,116,117]. SP, the undecapeptide of tachykinin, is known to stimulate human fibroblast migration [118], and can be released during a stress [119]. *S. epidermidis* is sensitive to SP via thermo unstable ribosomal elongation factor (EfTu) binding protein, which serves as a SP receptor [120]. EfTu is now recognized as a moonlighting multifunctional protein with ribosomal level function, exports to the bacterial surface and functions as an environmental sensor [121]. SP leads to a massive increase in *S. epidermidis* virulence and biofilm formation activity, via the raise of its adhesion potential on cultured keratinocytes and reconstructed epithelial models [120]. CGRP is a 37 amino acids peptide belonging to the calcitonin superfamily [113], which binds to the DnaK chaperone protein of *S. epidermidis* and increases its cytotoxicity and virulence towards keratinocytes. The authors suggest that the DnaK chaperone protein could be a moonlighting protein like EfTu, but this will require new investigations. CGRP induces a rearrangement of the *S. epidermidis* surface, increases its hydrophobicity and reduces biofilm formation in static conditions. In addition, *S. epidermidis*, in contact with CGRP, induces the activation of an inflammatory reaction via the increase of IL-8 secretion [120,122]. SP and CGRP can be released simultaneously with antagonist activities on *S. epidermidis* [123], independently of AMP secretion by keratinocytes [120,122].

Catecholamines such as epinephrine and norepinephrine, small cyclic compounds derived from tyrosine, are respectively hormones and neurotransmitters, known as stress mediators. In a recent study, the effects of these two catecholamines were investigated on RT4 acneic and RT6 non-acneic *C. acnes* strains. Catecholamines at 10^−6^ M have no impact on *C. acnes* planktonic growth but epinephrine increases biofilm formation of both strains whereas norepinephrine increases only biofilm formation of the RT4 strain. Catecholamines do not increase the intrinsic cytotoxicity or inflammatory potential of *C. acnes* on sebocytes but stimulate its effect on sebocyte lipid synthesis. Borrel et al. (2019) suggest that catecholamines may play an intermediate role between *C. acnes* and acne via an increase in biofilm formation and sebum overproduction, through an interaction with a catecholamine protein receptor having partial homology to the QseC identified in *E. coli* [124].

The presence of NUPs, human neurohormones, in the skin microenvironment of *C. acnes* could condition its colonization of sebaceous glands [100,112]. NUPs, small peptides with an omega structure and disulfuric bonds, are classified in atrial, type B and type C natriuretic peptides, named ANP, BNP and CNP, respectively [125]. ANP and CNP, alone or in association, at 37 °C in anaerobic condition increase the generation time and inhibit biofilm formation activity, biofilm density and thickness of the acneic RT4 and RT5 *C. acnes* strains. However, at 33 °C, biofilm formation activity and sensitivity to NUPs differ with a weaker inhibitory effect towards the *C. acnes* strains, suggesting that *C. acnes* metabolism is temperature dependent. Biofilm formation of *S. epidermidis* regarding thickness and biomass is stimulated at 37 °C but inhibited at 33 °C by ANP and CNP. In mixed or binary biofilms composed of *C. acnes*–*S. aureus* and *S. epidermidis*–*S. aureus*, ANP and CNP increase the competitive advantages for *C. acnes* and *S. epidermidis* towards *S. aureus* with its growth inhibition. Both ANP and CNP can be considered as having a thermostat function towards the local temperature, in order for the skin to modulate bacterial development according to normal or inflammatory conditions [100,126].

The shift from commensal to opportunistic pathogen of *S. epidermidis* and *C. acnes* is conditioned by the skin microenvironment. Since the biofilm formation of these strains and its reinforcement occur under the influence of host stress mediators, such as SP, catecholamines and NUPs, these parameters could be specific targets for cosmetics ingredient development.

### 2.3. Skin Ageing and Photoexposition Linked to Staphylococcus epidermidis and Cutibacterium acnes Dysbiosis

Age, which is an endogenous factor, may impact skin microbiota diversity because significant differences in abundance were observed between young and old subjects [127,128]. Significantly lower abundances of *Cutibacterium* in the cheek, forearm and forehead microbiomes in the older group were related to a decrease in the sebum secretion level in these aged skins. A significant decrease in the abundance of *Staphylococcus* in the forearm was also observed in the older group compared to the younger group. Shibagaki et al. (2017) suggest that depletion of *Cutibacterium* and *Staphylococcus* commensals may reduce their skin benefits and contribute to skin ageing signs [128]. A recent study by Rouaud-Tinguely et al. (2018) showed skin microbiota dysbiosis of mature Caucasian women. Skin-swab samples of the foreheads of two age-groups of healthy Caucasian women (17 younger women, from 21 to 31 years old, and 17 older women, from 54 to 69 years old) were evaluated for their bacterial communities using 16S rRNA gene metagenomic sequencing. At the phylum level, an increase in Proteobacteria and a decrease of Actinobacteria, and at the genus level a significant increase in *Corynebacterium* (+84%) and decrease in *Cutibacterium* (−40%) relative abundance were observed in mature skins. Rouaud-Tinguely et al. (2018) suggest that these dysbiosis could be explained by skin modifications such as lower pH, reduction of sebum secretion and modification of immune defenses, confirming the results of Shibagaki et al. (2017) for *Cutibacterium* [127,128]. In the study of Wang et al. (2012) [129], UV-B radiation was shown to decrease the production of *C. acnes*-derived porphyrins production in a dose-dependent manner and to increase its induced-apoptotic activity on melanocytes. However, *S. epidermidis* exhibited a protective effect on melanocyte development after UV exposure, which counterbalanced the negative effect of *C. acnes* [130]. This study showed an antagonism between *S. epidermidis* and *C. acnes* when exposed to UV.

Because of inter-individual and intra-individual variability, and multiple factors affecting skin microbiota, it is still not fully clear what healthy microbiota looks like [26]. However, it seems clear for the authors that the maintenance of the relative abundance of *S. epidermidis* and *C. acnes* and diversity might be a key factor in healthy skin conditions since they can turn to opportunistic pathogens in non-healthy ones [131]. Moreover, across this section, several objectives have emerged for cosmetic perspectives. The authors pointed out that acne prevention could be performed with the modulation of TLR2, the support for commensal growth of *S. epidermidis* and the inhibition of pathogenic *C. acnes* growth, biofilm and virulence factors. The prevention of pathogen invasion, such as *S. aureus*, could be executed via the promotion of commensal growth of *S. epidermidis* and *C. acnes* and metabolites production. The modulation of the skin microenvironment with the reduction of skin inflammation via host stress mediators, cytokines, or MMPs levels, which disturb its microbiota, are also considered. Since skin microbiota knowledge is relatively new and still uncomplete, and that claims for cosmetic applications required scientific assessments, the evaluation methods from a cosmetics perspective are discussed in the next section.

## 3. Evaluation Methods of Skin Microbiota Targeting *Staphylococcus epidermidis* and *Cutibacterium acnes* from a Cosmetics Perspective

Nowadays, when targeting cosmetic applications, several evaluation methods are being developed, using in vitro, ex vivo and in vivo methods. In vitro and ex vivo approaches are used for the screening assessment of cosmetic ingredients activity while in vivo approach is for the assessment of cosmetic products containing active ingredients. Cosmetic ingredients activity can be evaluated by targeting commensal or pathogenic strains of bacteria, such as *S. epidermidis* or *C. acnes*, in order to assess their growth kinetic, cell adhesion, cell invasion, biofilm production, cytotoxicity, virulence and quorum sensing. In vitro co-culture colonization models have to be developed for the assessment of (1) cosmetic ingredient activity on skin microbiota; and (2) the interaction between the microbiota and skin. Epidermal differentiation markers, such as filaggrin, integrin or keratin, and inflammatory markers, such as TLRs, AMPs or cytokines, are important targets for skin evaluations related to microbiota. Recent studies have developed the ability to integrate bacteria, yeasts or even full skin microbiome on monoculture or 3D skin models in order to evaluate cosmetic ingredients. New in vitro and ex vivo models of cosmetic evaluations have to integrate unselected skin microbiota, commensal and/or pathogenic microorganisms collected from a human donor in a 3D skin model supplemented with immune cells. In vivo cosmetic product application, in clinal studies, is followed by the evaluation of the full or partial skin microbiome in terms of bacterial diversity and relative abundance. These microbiome analyses are often completed with common in vivo properties evaluation, such as hydration, skin redness, elasticity, microrelief and sebum level, and only correlations are made.

### 3.1. Cutaneous Microbiome and Microbiota Evaluation

Bacteria are for now the main focus of the study design of skin microbiota. Bacteria are commonly identified and classified according to their small subunit 16S ribosomal RNA gene (16S rRNA), comprising conserved and variable regions. This gene is ubiquitous and highly conserved between different species [132]. In vivo cosmetic evaluation methods rely on cosmetic product application on human subjects followed by the evaluation of bacterial diversity. In vitro analysis of active ingredients without any cutaneous system mainly targets bacteria physiology with monoculture or coculture strains.

Clinical tests appear to be the most realistic and accurate method for final cosmetic product evaluation, since samples are directly taken from panelists. However, it is time and money consuming, as well as complex with the requirement of big data analysis in metagenomic, metaproteomic and/or metabolomic investigations. Skin sampling from individuals is performed by various techniques: non-invasive by surface scraping, skin swabbing and tape-stripping, or invasive by punch biopsies [133,134]. The most used sampling method is skin swabbing with a pre-moistened swab with 0.9% sodium chloride and 0.1% Tween-20. Swabbing results depend on the conditions of the sampling, such as number of strokes and the pressure applied while rubbing. ESwab^TM^, a flocked one, in the study of Van Horn et al. (2008), appeared to be the most efficient swab test [135]. The tape-stripping method, which uses adhesive tape, collects more viable bacteria from the stratum corneum and deeper layers in anaerobic areas than the swabbing one [136,137]. Skin biopsies are commonly used when analyzing hair follicles and collecting superficial aerobic microbiota and deep skin anaerobic microbiota [84]. A recent study using 16S rRNA gene sequencing demonstrated differences in both the diversity and taxonomic composition between the microbiome observed from non-inflammatory skin swabs and biopsies in 16 patients [138]. Choice of sampling method is not the only factor to impact cutaneous microbiome/microbiota studies because of internal and external variability. Indeed, many other factors should be included in the study, such as gender, drug treatment with antibiotic use, age, diet, ethnic origin, season, etc., and the body area where the sampling was performed.

After collection, sample processing is performed by DNA extraction with firstly microbial cell lysis, and then purification and characterization. The DNA is then amplified by PCR for the 16S rRNA + analysis region (V1–V3) [133]. qPCR and more precisely a PCR with propidium monoazide discriminates between live and dead bacteria, which allows to only quantify live collected bacteria [139]. The analysis continues with sequencing and comparisons with a database [133]. With DNA sequencing, it is common to identify dead bacteria, non-metabolically active bacteria or free DNA [4]. However, a new metatranscriptomic technique, known as mRNA sequencing, focuses only on metabolically active bacteria [140]. Furthermore, whole genome sequencing (WGS), also called shotgun metagenomic sequencing or NGS, analyzes the entire genetic material of a sample without targeting a specific region. Combination of 16S rRNA gene and metagenomic shotgun analysis allows the identification of a bacterial community at the genus level, at taxonomic species level and sometimes even at the strain level [141].

The study of bacteria physiology, namely, growth kinetics, cell adhesion and invasion, biofilm formation and production, cytotoxicity, virulence and quorum sensing, is generally performed using the cultural Pasteurian method. Bacteria physiology can be assessed on microplate cultures with a thermo-controlled multifocal reader [103]. This method is relatively simple and cheap. However, the media culture used differs from the skin surface environment and this technique identifies only 0.1 to 1% of the bacteria population, which underestimates the microbial community diversity and highlights only culturable bacteria with more than 90% G+ bacteria [142,143].

The in vitro models to study biofilm formation are numerous. The in vitro models commonly used are microtiter plate-based (MTP); flow displacement, such as modified Robbins device; PFR systems and centers for disease control biofilm reactors; and cell-culture-based and microfluidic devices. The MTP system is the most frequently used for biofilm growth on the bottom, walls or on the surface of a coupon of microtiter plate (6, 12, 24 or 96- wells). In vitro, biofilms using a single species with *S. epidermidis* or *C. acnes* [63] and with mixed-species biofilms of *S. aureus*–*C. acnes* and *S. aureus*–*S. epidermidis* [100,126] have been studied. Cell-culture-based and 3D skin biofilm models are discussed in the following Section 3.2: “*Skin–Microbiota Interactions and Cosmetic Active Ingredients Evaluation*”. Common visualization techniques of biofilm are scanning electronic microscopy, confocal laser scanning microscopy and Gram-bacteria staining [144]. *C. acnes* can be directly visualized in vitro and in skin tissue by an immunofluorescence microscopy assay and fluorescence in situ hybridization [145]. However, visualization by microscopic methods of skin bacterial communities has to be developed [133]. Multiple methods for quantification in microtiter plates of the total biofilm mass, matrix biofilm and bacteria—both dead and live—are available. These methods are based on several assays such as crystal violet, Syto9 coupled or not with propidium iodide, fluorescein diacetate, resazurin, XTT and dimethyl methylene blue [146]. Non-destructive analytical techniques with Raman spectroscopy, FTIR spectromicroscopy or magnetic resonance imaging are developed for biofilm morphology and composition analysis [147,148]. In vivo biofilm models in vertebrate animals [147] are not applicable in cosmetic applications because of European Regulation 1223/2009 [149].

The enounced in vitro methods do not reveal the full complexity of the skin–microbiota interactions. Thus, a more complex system with skin cells must be employed, using in vitro and ex vivo methods.

### 3.2. Skin–Microbiota Interactions and Cosmetic Active Ingredients Evaluation

Modulation of the skin microbiota composition is an innovative strategy in cosmetics in order to maintain or restore microbial homeostasis in case of dysbiosis and thus to exhibit a healthy skin microbiota balance. Assessment of microbial balance and its diversity in the presence of active ingredients is usually done in vitro on mono- or co-cultures of bacteria by growth measurement with bacteria counting [37], or relative growth quantified by qPCR [150]. These bacteria are from strain libraries or directly harvested from human skin. The use of microorganisms directly harvested on human skin allows easier work on the virulence of microorganisms, because of poor strain libraries in live bacteria [112].

In vitro studies of cutaneous cells–skin microbiota are based on the stratum corneum, cell monocultures such as keratinocytes, commonly HaCaT cells [119,122], and on sebocytes, or on 3D skin models [37,151]. Skin cells, such as keratinocytes and sebocytes, are in vitro infected by *C. acnes* and *S. epidermidis* [119,120,124,126]. The bacteria inoculation is commonly performed by first the recovery of the bacterial culture at the targeted studied phase or the culture supernatant if the focus is made on the metabolites secreted in the extracellular environment. Then, the bacterial culture is concentrated and resuspended in physiological water to get a multiplicity of infection (MOI) to infect the eukaryote cells in Dulbecco’s Modified Eagle’s Medium. The calculated MOI is bacteria-dependent, with 50 for *C. acnes* and 10 for *S. epidermidis* for instance. In general, the inoculation is done for 24 h and then cell viability can be assessed by MTT, WST-1 or LDH assays, or other assays can be performed such as immunoanalysis [150]. Skin markers of inflammation, such as AMP with hBD2 and hBD3, pro-inflammatory cytokines with interleukins and TNF-α and surface skin cell receptors with TLR2 and TLR4, can be estimated by immunochemistry using an enzyme-linked immunosorbent assay or by immunoblotting, such as Western blot, as well as immunostaining after cell culture inoculation with bacteria.

Studies are now generally conducted on 3D skin models, which are more relevant for cosmetic purposes and include from the simplest to the most complex: differentiated epidermis, skin equivalent, reconstructed human epidermis (RHE) and skin explants (ex vivo) [152]. The latest have the advantage to simulate the interactions between skin and bacteria. The skin equivalent corresponds to keratinocytes, derived from foreskin seeded onto collagenous, fibrin or glycosaminoglycan–chitosan matrices with fibroblasts or dermal substitutes derived from human fibroblasts [153]. The most used skin equivalent is the Labskin or Leed skin model, a 3D full-thickness living skin equivalent model with an air-exposed, well-differentiated epidermis. The 3D skin equivalent models or RHE such as Episkin^®^ were in several studies successfully inoculated with aerobic *S. epidermidis* and/or anaerobic *C. acnes* [123,153,154,155]. Skin equivalent models support their growth and allows the monitoring of these commensals. The skin equivalent and RHE are ideal for cosmetic studies since these models are “realistic” for the skin microenvironment. However, these artificial skin models do not have the total physiological response and thus do not reflect all the diversity and stability of skin microbiota when inoculated with specific bacteria. In addition, one of the limits of these models is that in major studies they included only a few bacteria with the two main bacteria of interest being *S. epidermidis* and *C. acnes*. However, new 3D skin models are now including “unselected” skin microbiota containing bacteria, viruses, yeasts and fungi, directly harvested from human subject, especially the cheeks or inner forearm. This unselected microbiota applied on 3D skin surfaces also supports the growth, microcolony and biofilm formation of *S. epidermidis* and *C. acnes* [156,157]. However, a difference is observed since the skin microbiota grows slower than the unique bacterium *S. epidermidis*, but stays stable and remains equivalent in density to that found on a skin donor [157].

Recent 3D skin models have incorporated immune cells to study the influence of commensal bacteria on the activation of immune dermal cells [158]. For example, activated CD4^+^ T-cells-supplementing a 3D skin model was used to investigate the responses of keratinocytes and fibroblasts after infection with *Candida albicans* yeast [159]. The authors suggest that similar studies could be performed with *S. epidermidis* and *C. acnes*.

All these in vitro, ex vivo or in vivo methods are useful and often combined by the cosmetic industry in their new products/ingredients development. However, all these techniques require further development to mimic the complexity of the in vivo condition and allow the description of the mechanistic pathway involved after cosmetic application.

## 4. Influence of Cosmetics on Skin Microbiota, Particularly on *Staphylococcus epidermidis* and *Cutibacterium acnes*

Intra- and inter-individual multiplicities in microbiota and chemical composition highlighted in the study of Bouslimani et al. (2015) are a huge challenge for the cosmetics industry and skincare development. Molecules associated with skincare or hygiene products last on the skin after their use despite several washings and these products might alter molecular and bacterial diversity [20].

Cosmetic ingredients used that are either functional ingredients, such as preservatives, oils and emulsifiers, or active ingredients, impact the skin microbiota and require attention. Indeed, conventional skincare or hygiene products such as soap, gel and cream contain preservatives and natural and synthetic chemicals that impact microbiota even if these effects are not fully investigated in detail for now [26,160]. Preservatives, such as phenoxyethanol, parabens, and methylisothiazolinone, are known to inhibit the survival of skin commensal bacteria such as *S. epidermidis* [161]. This alteration phenomenon depends on the residual activity of the preservative in the cutaneous environment [160]. Functional ingredients like oils, emulsifiers, fatty acids, gelling agents, thickeners and basic cosmetics for skincare, which both improve skin hydration, modulate microbiota diversity [162,163,164,165,166].

Dermo-cosmetic companies must undergo studies on skin microbiota while developing new ingredients or products in order to ensure consumers that their products maintain, improve a healthy microbiome, or restore a healthy skin-microbiome balance in case of a disturbed microbiome [167,168,169]. In 2019, the first certification of being “Microbiome-friendly”, set up by “MyMicrobiome”, appeared for a final cosmetic product. This certification is to validate that the product is contamination-free, that specific bacteria of the targeted area will be unharmed, that the microbiome diversity is preserved and that the skin balance is not disturbed (not by the suppression of commensals nor by the stimulation of pathogenic bacteria). For instance, a study in 2018 evaluated three different face washes, two “everyday” products and one 100% natural product, applied on 32 women’s upper volar forearm, for their effects on skin microbiome diversity, along with skin pH, moisture and trans-epidermal water loss (TEWL), washing twice a day for 4 weeks. Volunteers were divided into three groups according to skin characteristics: skin pH acid/normal/alkaline, very dry/dry/moist skin and very healthy/healthy/normal/stressed/critical skins, and each was assigned one product. All groups exhibited an increase in alpha diversity (species richness via operational taxonomic unit count and species diversity via the Shannon index) over time and their skin moved to a “healthier” state. The study suggested that synthetic ingredients modified the microbiota diversity, especially within the first two weeks [166]. A more recent study (2019) tested the ability of *S. epidermidis* to metabolize the functional ingredients commonly found in dermo-cosmetics formulae with native oils and waxes, fatty acid esters, fatty acid alcohols, fatty alcohols ethers, fatty acids and other [164]. The tested substances were formulated in water-in-oil and oil-in-water emulsions at concentrations commonly used in cosmetics—10% for oils, 5% for emulsifiers and 2% for fatty acids; the gelling agents and thickeners did not affect the in vitro bacterial growth of commensal *S. epidermidis* [164], and thus are microbiota friendly for this commensal. Future investigations should be performed to determine the effect of functional ingredients on skin microbiota diversity.

The cosmetic industry targets the potential of skin microbiota with more and more studies conducted on the research of active ingredients targeting skin microbiota and the assessment of their action mode: skin benefits, microbiota balance and bacteria physiology, such as induction or suppression of metabolic pathways, adhesion, biofilm formation, growth kinetics, virulence factors, quorum sensing, etc. [1,37,170]. Active cosmetic ingredients that target skin microbiota can be classified into the following categories:active ingredients, algal- or plant-based, and thermal water-based, which are not a nutrient source for microorganism;prebiotics: nutrients that confer a health benefit with modulation of structure and functionality of the host microbiota in topical application for the cosmetic sector [26]. Cosmetic prebiotic approaches are to maintain healthy skin microbiota, or improve the skin microbiota composition by limiting or reducing pathogen growth and in the same time preserve or stimulate commensal bacteria growth [143,156,171,172];probiotics: fragmented bacteria that confer health benefits to the host. Cosmetic products with “probiotics” or “probiotic ingredients” often contain non-viable bacteria, products of bacterial fermentation or cell lysates, which do not require changes in the preservative ingredient system [26]. Nevertheless, cosmetic products containing fragments of microorganisms as probiotics require care regarding safe production. For now, a strict definition of a probiotic in cosmetic products has not been established and these products should only follow European Cosmetic Regulation 1223/2009 [149];post-biotics: bacterial metabolites and or cell wall components released by probiotic microorganisms [160].

The major identified applications of active cosmetic ingredients targeting particularly *S. epidermidis* and *C. acnes* are (1) promotion of commensal metabolism and/or bacterial diversity with the ratio *S. epidermidis*/*C. acnes* for limitation of pathogen invasion; (2) reduction of pathogen growth, virulence and biofilms; and (3) modulation of the skin microenvironment and modulation of the immune responses since the cause–consequence link between the skin pathologies and microbiota dysbiosis is not yet established. These claims in dermo-cosmetics are based on the evaluation methods enounced in the previous Section 3: “Evaluation Methods of Skin Microbiota Targeting *Staphylococcus epidermidis* and *Cutibacterium acnes* from a Cosmetics Perspective”.

### 4.1. Promotion of Commensal Metabolism for Prevention of Pathogen Growth

Probiotics, mainly lactic acid bacteria from the *Lactobacillus* genus, are common bacteria of the intestinal microbiota used in food and dietary supplements targeting the intestine [173], and are known to promote commensal *S. epidermidis* growth. RELIPIDIUM^®^ by BASF Beauty Care Solutions France, yeast hydrolysate bio-fermented by a *Lactobacillus* strain, is an example of a probiotic that in vivo relatively increases commensal *S. epidermidis* growth on normal skin [174].

A recent clinical study investigated topical application of probiotic solutions from donor microbiomes on healthy volunteers’ areas rich in sebaceous glands. Two solutions correspond to a complete microbiome and three to defined sets of *C. acnes*, type IA and B, alone or in mixture. After 3 days of application, quantitative and qualitative increases in similarity between the donor and recipient microbiomes were observed even though significant variability among the recipient areas was also detected. This study demonstrates that the human skin microbiota composition can be modulated by application of *C. acnes* type IB with the positive features isolated from healthy individuals. However, the applied dose determines the duration of the applied strain’s abundance since abundance decreases gradually after final application [175].

Most of the active ingredients that promote commensal growth, such as *S. epidermidis*, also belong to the class of prebiotic, mainly poly- or oligosaccharides. Ecoskin^®^, by SOLABIA EUROPE, is a complex of pre- and probiotic with alpha-glucooligosaccharides obtained by enzymatic synthesis from alpha-glucan oligosaccharides, beta-fructooligosaccharides obtained by cold pressure of the *Polymnia sonchifolia* root, and *Lactobacillus* inactivated by tyndallization and freeze-dried. This active ingredient is suggested to promote skin microbiota equilibrium by supplying skin commensals in the substrate [176]. The BioEcolia^®^ prebiotic by SOLABIA EUROPE, α-glucan oligosaccharide obtained by glucosyltransferase enzymatic synthesis from saccharose and maltose, promotes commensal *S. epidermidis* MFP04 growth at 0.1, 0.5 and 1%, and increases its biofilm formation (+209% and +151% at 0.5 and 1%, respectively) and marginally rises the cytotoxicity of *S. epidermidis* on HaCaT keratinocytes at 1% (see Table 1) [170]. This result implies that BioEcolia^®^ at 0.1% is an interesting prebiotic that promotes commensal *S. epidermidis* growth without enhancing biofilm formation activity. Prebiotic OLIGOLIN^®^, arabinoxylan rich-oligosaccharides from flaxseed with a molecular weight between 5 and 15 kDa [177], and PROTEASYL^®^, peptidic purified pea extract, by BASF Beauty Care Solutions France, stimulates commensal strain *S. epidermidis* (ATCC 12228) growth and the production of lactic acid by lactic acid bacteria [178,179]. Seaweeds and their richness in oligosaccharides are potential prebiotics for commensal bacteria and thus for skin microbiota integrity [180]. For example, active ACTIBIOME^®^ by CODIF, a mixture of extracts from the brown seaweed *Laminaria digitata*, green microalgae *Chlorella vulgaris*, marine exopolysaccharides solutions and earth marine water, reverses microbiome dysbiosis linked to stress (65% against 35% with placebo) and reinforces microbial diversity. Skin disorders linked to this stress, such as skin pH, redness and non-uniformity of complexion, are also rebalanced. Extract composition promotes bacterial growth via nutrient source and protection against desiccation with exopolysaccharides [181]. Active EPS SEAPUR^®^ by CODIF, an exopolysaccharide solution from fermentation of marine planktonic microorganism, considered a post-biotic, is suggested to re-equilibrate the *S. epidermidis* and *C. acnes* ratio, and decrease the inflammation induced by bacterial stress [182]. A study conducted in 2005 on 523 women and 483 men revealed that 59% of women and 44% of men declared to have sensitive skin [183], subjective skin hyper-reactivity to various environmental factors characterized by an overheating feeling, tingling, itching and redness [184,185], but not associated to any dysbiosis of aerobic cultivable bacteria [186]. ExpoZen^®^ by GREENTECH, a *Halymenia durvillei* extract enriched in low molecular weight polysaccharides, was evaluated on samples from 30 volunteers with reactive and sensitive skins. After 28 days, the active ingredient increases the bacterial diversity and *S. epidermidis* level (see Table 1) [187].

Other active ingredients that are not classified as prebiotic/probiotic/post-biotic also promote commensal *S. epidermidis* metabolism, such as Viniderm^®^ and PS291^®^ (also called Téflose^®^). PS291^®^ is a polysaccharide obtained by bacterial fermentation, with a branched structure containing 60% rhamnose, glucose and glucuronic acid, by SOLABIA EUROPE. Both actives had no effect on the cytotoxicity of *S. epidermidis* on HaCaT keratinocytes, which is relevant for cosmetic applications. Viniderm^®^ increases *S. epidermidis* commensal MFP04 growth while PS291^®^ has no effect. PS291^®^ at all the tested concentrations (0.1, 0.25, 0.5 and 1%) and Viniderm^®^ at 0.1% decreases *S. epidermidis* biofilm formation (see Table 1) [170].

Modulation of skin microbiota with promotion of commensal growth also consists of the prevention of pathogen invasion, such as *S. aureus*, through production of organic acids that lower the surrounding pH environment, and through antimicrobial substance production, such as bacteriocins and bacteriocin-like substances. Commensal *S. epidermidis* appears to be a potential fighter against pathogen growth, especially *S. aureus*, which is of interest for the cosmetic sector in dysbiosis prevention. A study showed that the commensal *S. epidermidis* strain, collected from human subjects, cultured for proliferation and continuously applied in vivo on the subject’s face, allowed the increase of its skin colonization level. This was correlated with the rise in skin lipid content, limitation of water loss and improvement in skin moisture retention. *S. epidermidis* produces metabolites such as glycerine or organic acids like lactic acid and natural moisturizing factor, known to improve water retention, maintain a slightly acid condition on the skin surface and thus limit *S. aureus* colonization [188]. In a similar way via organic acid production, lactic acid bacteria have antimicrobial activity against skin pathogens [189]. Another study not conforming to cosmetic regulation, because of animal testing, exhibited interesting results in dysbiosis prevention but this will require new investigations to comply with dermo-cosmetic purposes. In this study, topical probiotic application of commensal *S. epidermidis* to mice, for treatment of AD, reduced the skin colonization by *S. aureus* through action of AMPs strain-specific to kill *S. aureus* and synergized with human AMP cathelicidin LL-37 [190]. Uriage^TM^ Thermal Water (UTW^®^), highly mineralized water with 11 g/L of dry residues characterized by its composition of sodium chloride sulfide, decreases *S. epidermidis* growth in a dose-dependent manner, but significantly increases at 50% its biofilm ability formation by +130% (see Table 1) [170]. We hypothesized this to be linked to the salt concentration of UTW^®^ promoting the production of PIA and poly-γ-glutamic acid [191]. The use of UTW^®^ in cosmetics is interesting despite its stimulation of the *S. epidermidis* commensal biofilm since its virulence is less than the one of acneic *C. acnes*, and that it exerts activity against this pathogen strain.

After promotion of commensal growth or metabolism via cosmetic prebiotic, probiotic or other active ingredients for pathogen prevention invasion, another important focus in cosmetic skin microbiota area is the reduction of pathogen metabolism related to skin disease such as *C. acnes* in acne vulgaris.

**Table 1 microorganisms-08-01752-t001:** Assessment of the effect of cosmetic active ingredients on *Staphylococcus epidermidis* and *Cutibacterium acnes* (growth, virulence, cytotoxicity and biofilm formation).

Active Names and Composition	*Staphylococcus epidermidis*				*Cutibacterium acnes*				References
	Growth	Virulence	Cytotoxicity on HaCat Keratinocytes	Biofilm Formation	Growth	Virulence	Cytotoxicity on HaCat Keratinocytes	Biofilm Formation	
BioEcolia^®^ Oligosaccharide with saccharose and maltose bond in α-1-2 and α-1-6	commensal strain MFP04				ns				[170,192,193]
	+	ns	+	+					
PS291^®^ Polysaccharide rich in rhamnose	commensal strain MFP04				normal skin strain and/or acneic strains (RT4: (HL045PA1/HM-516) and (RT5: HL043PA2/HM-514)				
	0	0	0	−	0	0	0	−	
ExpoZen^®^ Low molecular weight polysaccharides enriched in galactose produced by radical hydrolysis from *Halymenia durvillei*	+	ns			ns				[187,194]
Uriage^TM^ Thermal Water (UTW) Enrich in natural minerals 11 g/L (sulfates, chloride, sodium, bicarbonate, calcium, magnesium, potassium and silicon) and trace elements (zinc, manganese, cupper and iron)	commensal strain MFP04				acneic strains RT4 (HL045PA1/HM-516) and RT5 (HL043PA2/HM-514)				[103,170,192]
	− (idd)	ns	+	+	−	0	0	−	
Viniderm^®^ Rich in polyphenol and δ-viniferine	commensal strain MFP04				ns				[170]
	+	ns	0	−					
MPA-Regul^TM^ Vegetal polysaccharide rich in gluconic acid (obtained from enzymatic process) with UTW^TM^	ns				acneic strains RT4 and RT5				[103]
					0	0	0	− (idd)	
Myrtacine^®^ Lipophilic extract from *M. communis* leaves					strain CIP 53117T				[195]
					−	ns	ns	−	
BGM Complex Bakuchiol, *Gingko biloba* extract, and mannitol					strain CIP A 179				[196]
					−	ns			
ACNILYS^®^ *Rhodomyrtus tomentosa* berry extract					−	ns			[197,198]
DIOLÉNYL^®^ Ester of diol and polyunsaturated fatty acid					strain ATCC6919				[199]
					−	ns			

ns: not studied; 0: no effect; +: increased effect; −: decreased effect; idd: inhibition dose-dependent.

### 4.2. Reduction in Pathogen Growth, Biofilm Formation or Virulence: Example of Acne Dysbiosis, Frontier with Dermatology

In the case of acne, pathogenic *C. acnes* strains are the main targets in cosmetics for prevention of this skin pathology. Cosmetics companies have clearly identified this target and studies are being conducted with several objectives: (1) reduction of its virulence factors and modulation of cell cytotoxicity; (2) antibacterial activity against planktonic strains; (3) promotion of phylotypes diversity; and (4) inhibition of biofilm formation and maturation. It is noteworthy that some of the studies presented below do not target specific virulent *C. acnes* strains but the reference strains that can be discussed in the case of acne. Indeed, future studies must take into account the loss of phylotype diversity (predominance of phylotype IA1) and virulent *C. acnes* strains in order to get closer to “reality”.

#### 4.2.1. Decrease in Virulence Factors

Lipase production and activity of *C. acnes*, key virulence factor secreted by the bacteria that catalyzes the release of FFAs from sebum triglycerides, which trigger inflammation and keratinocyte overproliferation, is a common target of cosmetic actives.

BETAPUR^®^, Boldo, *Peumus boldus* Molina leaves extract with boldine ((S)-2,9-dihydroxy-1,10- dimethoxiaporphine) and catechin ((2S,3R)-2-(3,4-dihydroxy-phenyl)-3,4-dihydro-1(2H)-benzopyran3,5,7-triol) as the main components [200], and BIX’ACTIV^®^, *Bixa Orellana* seed extract, by BASF Beauty Care Solutions France (Lyon, France) inhibit *C. acnes* (ATCC 6919) strain lipase activity [201,202]. ACNILYS^®^, *Rhodomyrtus tomentosa* berry extract, by GREENTECH contains rhodomyrthone ([6,8-dihydroxy-2,2,4,4-tetramethyl-7-(3-methyl-1-oxobutyl)-9-(2-methylpropyl)-4,9-dihydro-1H-xanthene-1,3(2H)-di-one], a molecule known to inhibit lipase production [203]. In vivo, a topical cream containing BGM, a complex of bakuchiol, *Gingko biloba* extract, mannitol and zinc gluconate, made by the NAOS Group (Aix-en-Provence, France), applied twice daily for 84 days decreased significantly at 56 days the porphyrins parameters on the face of patients with oily skin and mild to moderate acne [196].

It is fundamental when targeting the acneic strain *C. acnes* to evaluate the impact of the active cosmetic ingredients on the virulence and cell induced-cytotoxicity of the bacteria. UTW^®^ and MPA-Regul^TM^ (complex of gluconic acid-rich polysaccharide with UTW^®^) by URIAGE and PS291^®^ have no impact on the virulence of *C. acnes* acneic RT4 and RT5 strains and thus present no induced cytotoxicity towards keratinocytes (see Table 1) [103,204]. BIX’ACTIV^®^ decreases the in vitro *C. acnes* (ATCC 6919) virulence towards sebocytes [201].

#### 4.2.2. Antibacterial Activity

Another cosmetic target in acne prevention is antibacterial activity towards *C. acnes* at the planktonic state.

UTW^®^ at 50% alone impacts the growth kinetics of planktonic acneic strains by decreasing the generation time of the acneic RT4 and RT5 *C. acnes* strains and the final biomass, while PS291^®^ alone has no effect on their growth. In combination, UTW^®^ and PS291^®^ at 50% and 4%, respectively, have a similar effect on *C. acnes* growth than UTW^®^ alone, which suggests that *C. acnes* does not metabolize PS291^®^ (see Table 1) [192]. Dermo-cosmetic Effaclar^®^ Duo+ by La Roche-Posay Laboratoire Dermatologique (Asnières, France), contains lipohydroxy acid, salicylic acid, linoleic acid, niacinamide, piroctone olamine, a ceramide and thermal spring water. This product was applied in vivo daily for 28 days on half of the face of 26 subjects with mild to moderate acne. Lipoxydroxy acid is commonly used in acne skin care [205]. In the study, the number of *Staphylococcus* spp. and Actinobacteria, including *Corynebacterium* and *Cutibacterium*, was decreased after 28 days, and thus Effaclar^®^ exhibited antibacterial activity [62]. Diolenyl^®^ by PIERRE FABRE, esters of alkane diol and PUFAs such as omega 3 or 6, undergoes cleavage by *C. acnes* lipases. Released diol and PUFAs exert, respectively, antibacterial and anti-inflammatory activities towards *C. acnes* colonization (see Table 1) [199]. Antibacterial activity of BGM was quantified towards the reference *C. acnes* strain (CIP A 179) in vitro with the Minimum Inhibitory Concentration (MIC) growth up to 0.0005% for BGM against 0.12% for zinc gluconate (see Table 1).

A study screened 57 seaweed species for their potential antibacterial activity against reference *C. acnes* (KCTC 3314) [206]. Two aqueous fractions, after methanol extraction, from the Phaeophyta *Ecklonia cava* and *Ecklonia stolonifera* exhibited antimicrobial activity against *C. acnes* after a 72-h incubation with 5 mg/disk. The identified inhibition zones were 2.8 ± 1.0 mm and 2.5 ± 0.5 mm, respectively [206]. Another study screened 342 species of marine seaweed for antibacterial activity against *C. acnes* (ATCC 11827). Only 13 seaweed species, including red and brown ones, were found to have anti-*C. acnes* activity based on a disk inhibition assay. A methanol extract from the marine brown seaweed *Sargassum macrocarpum* exhibited a compound named sargafuran (C_27_H_36_O_4_) with a MIC of 15 μg/mL against *C. acnes* [207,208]. An ethyl acetate extract and dichloromethane fraction, enriched in glycolipids, with the main active glycolipid 2′,3′-propyl dilinolenate-β-D-galactopyranoside (C_45_H_74_O_10_), from the brown seaweed *Fucus evanescens* exhibited antibacterial activity against the reference and clinical isolates of *C. acnes* strains [209]. The anti-acne-related effects of the phlorotannins isolated from the brown seaweed *Eisenia bicyclis* by methanolic extraction and solvent fractionation, with hexane, dichloromethane, ethyl acetate and butanol in sequence, was investigated on the *C. acnes* reference strain (KCTC 3314) and clinical isolates (2875 and 2876). The methanol extract had potential antibacterial activity but the ethyl acetate fraction, and particularly fucofuroeckol-A, showed the highest antibacterial activity against the *C. acnes* strains [210].

Prebiotics and probiotics also have the potential for modulating acne in cosmetic applications. Prebiotics composed of konjac glucomannan hydrolysates in spray formulation at 5% (*w*/*v*) reduce acne [211] via the inhibition of *C. acnes* growth [212,213]. Topical probiotics using commensal *S. epidermidis* can be suitable in acne disease thanks to its ability to change in vivo the host microbiome and in vitro inhibit *C. acnes* growth [53,214]. Probiotics made of a microtube array membrane (MTAM) encapsulating live fermenting *S. epidermidis* and 2% glycerol, applied on *C. acnes*-injected mouse ears, but not conforming to cosmetic regulations because of being animal tested, reduced *C. acnes* growth [214]. The authors suggest that a probiotic containing *C. acnes* bacteriophages (viruses infecting *C. acnes*) or *C. acnes* bacteriophage endolysins, could be of interest to eliminate acne-associated *C. acnes* strains [215]. Indeed, *C. acnes* phages, with limited genetic diversity, exert a broad ability to kill *C. acnes* isolates from acneic comedones. This is performed via its endolysin protein that degrades the *C. acnes* peptidoglycan cell wall [216,217,218].

#### 4.2.3. Promotion of *Cutibacterium acnes* Phylotype Diversity

As previously discussed, acne is related to a loss of *C. acnes* phylotype diversity and an increase in phylotype IA1. The promotion of *C. acnes* phylotype diversity with a decrease in phylotype IA1 and increase of other phylotypes by an active cosmetic ingredient is really interesting for acne prevention and deserves more attention for future development.

*Rhodomyrtus tomentosa* fruit extract, an active molecule of ACNILYS^®^, tested at 2% with all *C. acnes* phylotypes (IA1, IA2, IB, IC, II and III), specifically modulates skin microbiota by increasing the *C. acnes* phylotypes II and III, and decreasing phylotype IA1 [219].

#### 4.2.4. Inhibition of Biofilm Formation and Maturation

As mentioned previously, the *C. acnes* biofilm, a key virulence factor of acne pathogenesis, is a frequent objective to inhibit in acne prevention for cosmetic issue.

In vitro, *C. acnes* RT4 and RT5 acneic strains, collected from the pilo-sebaceous follicles of acneic patients, were treated with MPA-Regul^TM^. The active ingredient induces the inhibition of *C. acnes* biofilm formation in a dose-dependent manner at 0.1, 0.5 and 1% without any modification of biofilm thickness [103]. PS291^®^ is able to induce the reduction of *C. acnes* adhesion and biofilm formation in a dose-dependent manner [193]. UTW^®^ and PS291^®^ have both antibiofilm activity on RT4 and RT5 acneic *C. acnes* strains, with a reduction in the biofilm thickness and density, as well as a reduction in the projection numbers and structural elements on the biofilm surface (see Table 1). However, UTW^®^ and PS291^®^ do not modify *C. acnes* metabolic activity and have a limited effect on the increase of its surface hydrophobicity [192].

Rhodomyrthone, active molecule of ACNILYS^®^, inhibits biofilm formation and disorganized the established biofilm of five *C. acnes* clinical biofilm-forming isolates and one reference strain (DMST 14916) (see Table 1) [203].

Myrtacine^®^ New Generation by PIERRE FABRE, a lipophilic extract from *Myrtus communis* leaves, was tested on reference *C. acnes* strains (CIP 53117T) at concentrations between 0.0001% and 0.1%, *w*/*v*. Myrtacine^®^ inhibits in static and dynamic conditions biofilm formation and mature biofilm of *C. acnes* (see Table 1) [195].

### 4.3. Modulation of the Skin Microenvironment and Immune Responses

The modulation of the skin microenvironment during dysbiosis conditions is a key feature for further active cosmetic ingredient research. In vitro studies were conducted on SP–*S. epidermidis* and catecholamines–*C. acnes* interactions after exposure to active cosmetic ingredients [119,124]. The UTW^®^ effect was investigated after exposure of RT4 acneic and RT6 non-acneic *C. acnes* strains to host stress mediators, epinephrine and norepinephrine. UTW^®^ at 30% partially inhibits the catecholamine-induced biofilm formation of the RT4 strain but stimulates epinephrine-induced biofilm formation of the RT6 strain. This results from the UTW^®^ intrinsic effect on the basal level of the biofilm production rather than a direct interaction with the action of the catecholamines and depends in vitro on the surface used for the biofilm growth [124]. After 1 h of exposure of HaCat keratinocytes to SP at 10^−6^ M, UTW^®^ and PS291^®^ were able to completely neutralize the increase in the SP-induced cytotoxic activity of *S. epidermidis* [119].

Sebum overproduction regulation is a way to modify the skin microenvironment in acne dysbiosis since sebum is the main nutrient source of *C. acnes*. BETAPUR^®^ and BIX’ACTIV^®^ are examples of cosmetic actives that decrease the overproduction of sebum by sebaceous glands [201,202].

The modulation of immune responses and skin barrier function is of interest when targeting skin microbiota wellness. Modulation of immune responses can be done via stimulation of hBD production, synthesis of TLRs, cytokines regulation and strengthening of the skin barrier function via ceramide, collagen, elastin, hyaluronic acid production, proliferation ability of skin cells or MMP inhibition [220]. Ingredients using probiotic strains promote skin barrier function reinforcement, and exhibit anti-inflammatory and moisturizing properties without a negative impact on skin microbiota diversity [221,222]. For instance, post-biotic LACTOBIOTYL^®^ by SILAB, a cyclic polyols pinitol type produced from jojoba fermentation by *Lactobacillus arizonensis*, is suggested to respect microbiota equilibrium, to improve skin barrier integrity by cohesion and structure of the protein and lipidic markers and to accelerate epidermal renewal [222]. ECOBIOTYS^®^ by SILAB, a yeast extract enriched in biopeptides from *Metschnikowia reukaufii* isolated from the nectar flower *Hoya carnosa*, specifically re-equilibrates the microbiota of mature skin and restores the function of the immune skin barrier [223]. A *Lactobacillus pentosus* extract, the active ingredient of Phytobioactive BIOTILYS^®^ by GREENTECH, after 6 days of treatment at 2% on an ex vivo skin model significantly increases (by 229%) hBD2 expression and decreases (by 57%) TLR2 expression.

Immune responses modulation is often targeted in acne prevention. BETAPUR^®^ stimulates hBD3 mRNA expression without inducing cytokines synthesis [202]. TLR2-Regul^TM^, by URIAGE, a natural vegetal extract umbellifers with synthetic amphiphilic lipid C_18_H_39_NO_3_, increases hBD2 expression and thus highlights its antimicrobial property [103]. An ex vivo study of the active TLR2-Regul^TM^ applied on human skin explants in contact with acneic *C. acnes* strains also induced a decrease in IL-8, highlighting its anti-inflammatory property [103]. This anti-inflammatory property was confirmed with an in vivo study conducted on adult patients with light polymorph acne. The efficiency of two oil-in-water emulsions, A or B, applying one of the emulsions twice a day for two months, was assessed. Emulsion A was formulated with Hyséac 3-Regul^®^ cream by URIAGE + MPA-Regul™ 1% + Alpha-Hydroxy-Acids (AHA) + TLR2-Regul™ + UTW^®^, and emulsion B with dermo-cosmetic anti-acneic cream with AHA and soothing agents. Emulsion A induced a significant decrease in the inflammatory and retentional acneic lesions score after 28 days treatment compared to emulsion B [103]. The anti-inflammatory activity of BGM was shown ex vivo through a decrease of IL-8 and TNF-α by −45% and −46%, respectively, *p* < 0.01, on human skin explants, from a 40-year-old female subject exposed to lyophilized *C. acnes* strain, on which the cream containing a BGM complex was applied twice daily for 3 days.

## 5. Conclusions

In recent years, increasing numbers of works on the skin microbiome and microbiota have appeared. Indeed, the skin microbiota is primordial in skin homeostasis. Nevertheless, further investigations must be performed to gain a complete picture of the skin’s microbiome and microbiota as well as their complex interactions with the skin. Cosmetic industries that have underlined the sentinel role of *S. epidermidis* and *C. acnes* try to better understand the interaction of these two bacteria, which continuously interact with the skin system and its microenvironment. Cosmetic industries create or valorize active ingredients that maintain or restore a particular skin microbiota after external stress or skin modification. The cosmetic sector, directly involved in the preservation or restoration of skin homeostasis, have now the duty to consider skin microbiota. These investigations, especially in a cosmetic activities context, will require new or improved evaluation methods in order to reflect the complexity of the microbiota, skin and their interactions. The identified future challenges for the cosmetic industry are now to develop cosmetic functionals and actives ingredients and final products that are harmless to commensal microbiota and thus create a microbiota-friendly ingredient database. Furthermore, because the skin microbiota is unique to each individual, a new area of research will be the personalization of skincare products adapted to each skin ecosystem. Finally, pre- and probiotics are also pointed out regarding their systemic delivery (ingestion) from a cosmetic perspective, since the modulation of the gut microbiota causes beneficial effects in skin [143,224]. In this way, another area to consider will be the gut–brain–skin–microbiota axis as a new dimension of cosmetic innovation.

## Figures and Tables

**Figure 1 microorganisms-08-01752-f001:**
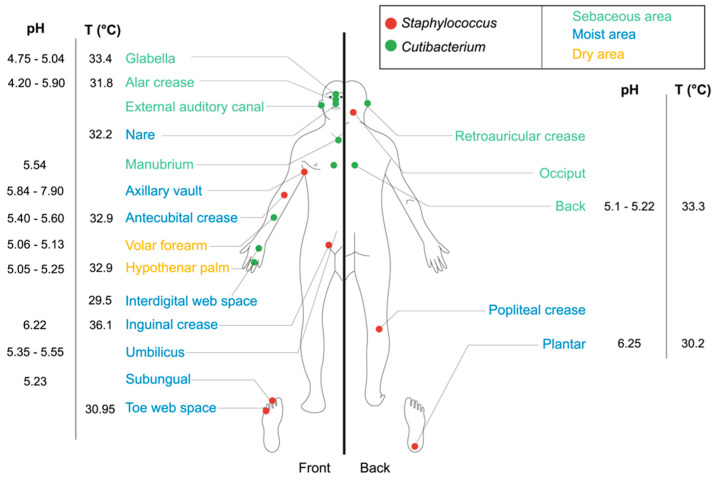
Topographical distribution of *Staphylococcus* and *Cutibacterium* bacteria, adapted from Wilson (2005), Grice et al. (2015) and Kong et al. (2012) [6,17,27].

**Figure 2 microorganisms-08-01752-f002:**
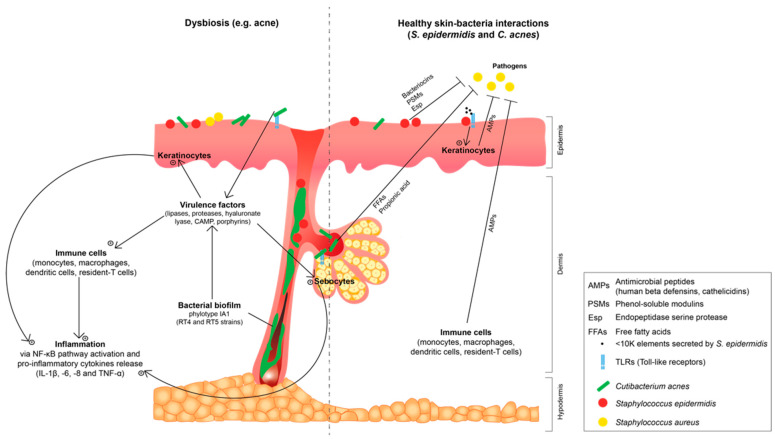
Skin–bacteria interactions in dysbiosis (e.g., acne) and healthy skin: focus on *Staphylococcus epidermidis* and *Cutibacterium acnes*, adapted from Claudel et al. (2019) and O’Neill et al. (2018) [44,45].

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
