# Peer review of "Staphylococcus epidermidis and Cutibacterium acnes: Two Major Sentinels of Skin Microbiota and the Influence of Cosmetics"

_microorganisms, 2020, doi:10.3390/microorganisms8111752_

Round 1

Reviewer 1 Report

Fournière et al provide an in-depth review on the role of Staphylococcus epidermidis and Cutibacterium acnes as sentinels of the skin microbiota.  They also report on the effect of pre- and probiotics as active compounds of cosmetics.

In general, this is an excellent review on an exciting new field in skin microbiota research, with satisfactory citation of  the relevant literature. It deals with the role of skin microbiota dysbiosis in two frequent skin disorders, atopic dermatitis and acne vulgaris. Both skin diseases are characterized by colonization with pathogenic bacteria – i.e. S. aureus and C. acnes, that are normally under control of non pathogenic microbiota.

It has been shown recently that the shift from commensal to opportunistic pathogen of S. epidermidis and C. acnes is conditioned by the skin microenvironment. Of importance, acne is not due to a significant increase of C. acnes but rather to a loss in C. acnes phylotype diversity.

As commensals S. epidermidis and C. acnes have a symbiotic or even mutualistic relationship with the cutaneous microbiota system. Consequently, a new strategy in the treatment of both atopic dermatitis and acne is to support the skin microbiome diversity rather than fighting pathogenic strains with topical antibiotics or antiseptics.

Modern actives in cosmetics / cosmeceuticals aim at maintaining a healthy environment by promoting the growth of commensal bacteria such as S. epidermidis. Several examples are given such as prebiotics, probiotics and various poly- or oligosaccharides from marine organisms and plant extracts. These actives increase bacterial diversity instead of aiming at killing pathogenic strains.

The whole topic stands for a paradigm shift in the understanding and therapeutic concepts of skin disease, away from killing the enemy towards strenthening the healthy environment and the skins self-regulatory capacity.

Reviewer 2 Report

The review of Fournière et al is (in parts) very broad and exhaustive, with an interesting application section (sections 5 and 6).

In my opinion, the review can be divided into two parts. Each part has a different level of quality.

Sections 1-3 (and to some extent section 4) contain basic knowledge about C. acnes and S. epidermidis and the skin microbiota in general. It is often weak and superficial, with many microbiological inaccuracies. A few examples below.

I would recommend to reduce/delete this part and refer to other reviews in this field. For example, one could reduce this first part in such a way that it contains only the essential information needed to understand sections 5-6.

Sections 5-6 are stronger; they contain results from applied studies, and I have not seen many reviews about the application perspective of skin microbiome research.

Some further comments:

Language: grammar needs major revision, throughout the text. Several wrong terms such as “metasequencing”

Many parts of the review are very general and superficial, e.g. Section 2.1 and 2.2

Many statements in the manuscript are hypotheses but not labeled as such, mentioned by the authors themselves, e.g. (lines 187ff): “In healthy skin microbiota, C. acnes is more stable than S. epidermidis due to its localization in pilosebaceous unit less subjected to environmental factors. However, no publication demonstrates this hypothesis for now.”

Another example: “Its bacteriocins and bacteriocin-like molecules production such as propionicin PLG-1, jenseniin G, propionicins SM1, SM2 T1, and acnecin, allow its follicle and skin surface colonization [29,69]”

This has never been shown in vivo, to my knowledge.

The whole text needs to be revised to carefully separate speculations and hypothesis, from real facts (that are scarce in this research field, in particular regarding C. acnes) or oversimplifications.

There are also several imprecision errors such as “The bacteria metabolizes fatty acids and other sebaceous fluids or lipids to generate through its lipase  activity free fatty acids (FFAs) such as propionic and acetic acids”. Lipases do not produce propionic acid and acetic acid. The latter SCFAs are produced via the core metabolism (wood-werkman cycle etc)

Some parts of the review are difficult to understand; I think grammar issues are a main reason. E.g. lines 212ff “In the study of Wang et al. (2012) [60], UV-B were shown to decrease in dose-dependent manner C. acnes porphyrins production and increase C. acnes apoptotic bacteria activity on melanocytes.”

What the probably mean: In the study of Wang et al. (2012) [60], UV-B radiation was shown to decrease the production of C. acnes-derived porphyrins in a dose-dependent manner. (The second part of their sentence I did not understand: “increase C. acnes apoptotic bacteria activity on melanocytes”). There are many examples like this throughout the text.

The review gets more interesting in sections 5 and 6. I would strongly advice to reduce this review to content that is of direct interest for these sections e.g. from chapter 5 onwards “(Evaluation methods of skin microbiota targeting Staphylococcus epidermidis and Cutibacterium acnes for cosmetic perspectives)”
